# SimiGrad: Fine-Grained Adaptive Batching for Large Scale Training using Gradient Similarity Measurement

**Heyang Qin**
University of Nevada, Reno
heyang_qin@nevada.unr.edu

**Samyam Rajbhandari**
Microsoft
samyamr@microsoft.com

**Olatunji Ruwase**
Microsoft
olruwase@microsoft.com

**Feng Yan**
University of Nevada, Reno
fyan@unr.edu

**Lei Yang**
University of Nevada, Reno
leiy@unr.edu

**Yuxiong He**
Microsoft
yuxhe@microsoft.com

## Abstract

Large scale training requires massive parallelism to finish the training within a reasonable amount of time. To support massive parallelism, large batch training is the key enabler but often at the cost of generalization performance. Existing works explore adaptive batching or hand-tuned static large batching, in order to strike a balance between the computational efficiency and the performance. However, these methods can provide only coarse-grained adaption (e.g., at a epoch level) due to the intrinsic expensive calculation or hand tuning requirements. In this paper, we propose a fully automated and lightweight adaptive batching methodology to enable fine-grained batch size adaption (e.g., at a mini-batch level) that can achieve state-of-the-art performance with record breaking batch sizes. The core component of our method is a lightweight yet efficient representation of the critical gradient noise information. We open-source the proposed methodology by providing a plugin tool that supports mainstream machine learning frameworks. Extensive evaluations on popular benchmarks (e.g., CIFAR10, ImageNet, and BERT-Large) demonstrate that the proposed methodology outperforms state-of-the-art methodologies using adaptive batching approaches or hand-tuned static strategies in both performance and batch size. Particularly, we achieve a new state-of-the-art batch size of 78k in BERT-Large pretraining with SQuAD score 90.69 compared to 90.58 reported in previous state-of-the-art with 59k batch size.

## 1 Introduction

Recent years have witnessed the success of deep learning in a variety of applications, such as computer vision [1] and natural language processing [2, 3]. To push the performance boundaries, deep learning models in many applications have become increasingly complicated, and training such models requires large amounts of data as well as high cost. Taking GPT3 [3] (an autoregressive language model that uses deep learning to produce human-like text) as an example, the state-of-the-art model has 175 billion parameters and is trained on 300 billion data, which costs $12 million for a single training run [4]. Therefore, it is of critical importance to study scalable and efficient training for large scale machine learning [5, 6].

The core idea of large scale machine learning is to parallelize model training across vast amounts of computing resources to speed up the training process. However, the overhead of the model synchronization due to massive parallelism in training can be the bottleneck for large scale training. To reduce the model synchronization overhead, large-batch training is a commonly employed strategy that can reduce the synchronization frequency. Taking BERT training as an example, increasing batch size from 256 to 4k can increase the training speed 2.5x faster on a 16-GPU cluster. OpenAI uses a batch size of 3.2 *million* to train GPT3 over a supercomputing cluster [3], which accelerates the training speed by approximately over 500x compared to sample-by-sample run on a 16-GPU cluster.

35th Conference on Neural Information Processing Systems (NeurIPS 2021).

Despite the benefits of large batch size, large batch size may degrade the model generalization performance [7, 8, 9]. To address this issue, existing works [10, 11, 12] demonstrate that a carefully tuned learning rate scheduling can help to mitigate such performance loss caused by large batch size. However, manually tuning learning rate requires significant amounts of efforts, which limits its potential use for wide adoption. [13, 14] develop automated learning rate tuning methods for large batch size, which, however, rely on users to set appropriate batch sizes. [15] develops adaptive batching using second order information, aiming to reduce the generalization impact due to large batch sizes. [16] uses the gradient variance information to guide the batch size and learning rate configurations. Due to the high cost of analyzing the gradient variance and the second order information, existing methods can adapt the batch size only at a coarse-grained level (i.e., at the end of each epoch). It is worth noting that within each epoch, the variance of gradient may change swiftly and significantly [17, 18] (also see Section 4.1). Therefore, the course-grained methods cannot find a suitable batch size to reduce the performance loss, and thus miss the opportunity to achieve larger batch sizes with less performance impact.

To address the above challenges, we propose a fine-grained batch size adaption approach SimiGrad to optimize the batch size at the iteration level. SimiGrad is driven by a novel gradient similarity measurement that can accurately capture the gradient variance information in a lightweight manner. We use the proposed gradient similarity measurement to derive the optimized batch size and learning rate with an automated algorithm. We integrate SimiGrad into mainstream machine learning frameworks by providing an open-sourced plugin tool.

We evaluate SimiGrad via extensive experiments using popular benchmarks, such as CIFAR10, ImageNet, and BERT-Large. The experimental results demonstrate that SimiGrad can consistently outperform state-of-the-art methods in both the average batch size and the model generalization performance. Particularly, we achieve a new state-of-the-art batch size of 78k in BERT-Large pre-training with the SQuAD score 90.69, compared to 90.58 reported in the state-of-the-art LAMB[19] with 59k batch size. We also achieve better Pareto frontier for the trade-off between the batch size and the model performance, compared with state-of-the-art methods.

We summarize our main contributions as follows:

- We develop an accurate gradient similarity measurement for capturing the gradient variance at the iteration level. This measurement can be directly inferred from the gradients of replicas with low computation overhead.
- We develop a fully automated fine-grained adaptive batching algorithm that can identify the optimal batch size and learning rate using the gradient similarity measurement.
- We integrate SimiGrad into mainstream machine learning frameworks and open-source it [1].
- SimiGrad enables a record breaking large batch size of 77k for BERT-Large pretraining. With very large batch sizes, SimiGrad is able to achieve better performance than state-of-the-art methods.

## 2 Background and Related Work

### 2.1 Mini-batching Training

The objective of training a neural network is to optimize the neural network parameters $\theta$ by minimizing the loss function $L(\theta)$. One typical method of training a neural network is the gradient descent method [20], which computes the gradient of $L(\theta)$ with respect to $\theta$ over the entire dataset; however, it could be too expensive for training deep neural networks with large training datasets. A typical solution is based on mini-batch stochastic gradient descent (SGD) [21], which samples a small subset, namely mini-batch, from the dataset and approximates the gradient using this batch instead. Such approximation introduces inevitable noise to the gradients. By increasing the batch size (i.e., the number of samples), we can reduce the noise level, but it is a diminishing return, meaning that after a sweet point we can gain almost no more information by further increasing the batch size [22]. In other words, the gradient computed on a too small batch is biased with noise, whereas a too large batch leads to a waste of time and insufficient updates to the model [23]. This makes the batch size a key control knob, and the choice of the batch size is of critical importance. In practice, a large batch may be too large to fit into GPU memory. The solution [24, 25] is to break it into many micro batches and accumulates the gradients of all micro batches. This method is called gradient accumulation, which is widely used in large scale training.

### 2.2 Large Batch Size Training

For large scale neural network training, it is desirable to have a large batch size, as it improves hardware utilization and system scalability as well as reduce wall clock training time. However, there

---

[1]Code repo is available at: https://github.com/HeyangQin/SimiGrad

is a dispute as to how exact large batch size would impact the model performance. In [7, 8], it is reported that training neural network model with a large batch size leads to a non-trivial generalization gap compared with a small batch size. [10] suggests that this gap could be closed by adjusting the training regime. [11] observes from a large number of training runs that this gap could be mitigated if the learning rate is well-tuned for a large batch size. [12] further observes that the range of "good" learning rates is shrinking as the batch size grows. [9] further argues that with carefully tuned hyper parameters, the smaller batch sizes still outperform large ones in terms of test accuracy.

Despite the disagreement over large batch size's generalization performance, much effort has been made on how to choose an appropriate batch size and the corresponding learning rate for neural network training. [26] formulates SGD as stochastic differential equation and suggests that there exists an optimal noise scale related to the batch size, the learning rate and the dataset. Along this line, [16] formulates the noise scale in a different way and tests it in a variety of applications. Another line of research explores the potential of large batch size training from a practical perspective by fine tuning the training setting, especially the learning rate. [24] suggests that the learning rate should be increased proportionally to the square root of the increase ratio of the batch size. [13] proposes the warmup learning rate scheduling that can scale up the batch size to 8k for ResNet-50 on ImageNet.

The most closely related works to this paper are AdaScale SGD [14] and gradient noise scale [16]. AdaScale SGD [14] proposes a novel learning rate scaling rule based on the gradient variance for training with a large batch size; however, its performance hinges on the choice of a suitable batch size, which is challenging to determine in practice. Gradient noise scale [16] proposes to predict the batch size by comparing the gradients before and after the allreduce call in the data parallelism as an alternative implementation. SimiGrad differs from gradient noise scale [16] in three ways: 1) the approach in [16] highly depends on the parallelism degree of the system, whereas SimiGrad supports flexible parallelism degree; 2) the measurement proposed in [16] is noisy and needs to use a hand-tuned exponential decay parameter to stabilize it; and 3) [16] assumes the learning rate is well-tuned and close enough to optimal, whereas SimiGrad has no assumptions on learning rate. In particular, by evaluating [16] in a common cluster of 16 GPUs, we observe a very high noise level. We also observe that the noise level decreases as the number of GPUs increases, which indicates that the design in [16] highly depends on the parallelism degree. In comparison, SimiGrad can derive relatively stable measurements in a parameter free and system independent manner as shown in Fig. 3.

### 2.3 Adaptive Batching

Using a fixed batch size throughout the training may not be the best practice. A desirable approach is to use adaptive batch sizes during training, which is suggested by research from both theoretical [22] and practical [27, 28] perspectives. [22] converts the existing learning rate schedule into the batch size adjustment schedule. Yet the optimal learning rate is difficult to obtain in practice and the batch size adjusted in this way has limited scalability. [27, 28] use fixed multi-step batch size schedulers that closely resemble the multi-step learning rate scheduler. Recent works focus on utilizing the gradient noise variance information to adjust batch sizes by various rules and measurements. [29] proposes a coupled criterion for adjusting both the batch size and the learning rate. [15] adjusts the batch size using the second order information and adversarial training. [16] proposes a gradient noise scale to adjust the batch size.

## 3 Motivation and Challenges

Due to the high cost of analyzing the gradient variance and the second order information, existing methods can adapt the batch size only at a coarse-grained level (i.e., at the end of each epoch). Our key observation is that the gradient variance may change swiftly across iterations within one epoch (see Fig. 2 in [17] and Fig. 3 in [18]), which cannot be captured by the existing methods. Taking [16] as an example, to get the gradient variance information, it needs to run at least 20 batches with different batch sizes. To perform such analysis at each iteration, the extra overhead due to gradient variance analysis would be more than 20 times of the original training, which is infeasible in practice. Due to these issues, the batch size usually needs to be compromised and large scale training methods that achieve state-of-the-art performance still heavily rely on manual tuning efforts [5, 6].

To address these issues, one promising solution is by improving the granularity of gradient analysis, which, however, is challenging due to the following reasons.

- To compute the gradient variance, we need to first collect the gradient information from batches at different sizes and then take large amounts of computing time and resources. For example, computing the gradient variance for a single 8k batch in BERT training takes 441ms and the time grows significantly with the batch size.

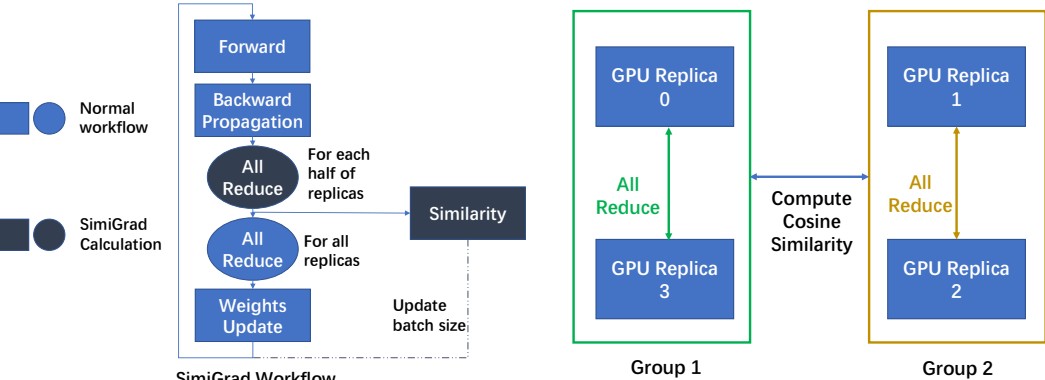

Figure 1: Workflow of how SimiGrad is built into normal data parallelism pipeline.

Figure 2: Cosine similarity computation from two groups of replicas.

- Even if we can obtain the gradient variance, it is not straightforward to derive the optimal batch size as the gradient covariance is a big matrix.
- Existing batch adaption methods usually require a fine-tuned learning rate scheduler to achieve a good performance. Fine tuning the learning rate at the iteration level requires significant amounts of manual efforts, which may not be feasible in practice.
- Due to the frequent gradient analysis, the system deployment is no longer a trivial engineering problem as the analysis and tuning overhead can disrupt the well optimized training pipeline in existing machine learning frameworks. For example in [16], executing multiple batches with different batch sizes to compute the gradient variance is not supported by the training pipeline and may lead to inefficient memory use.

## 4  SimiGrad: Fine-Grained Adaptive Batching Methodology

To address the above challenges, we propose a fine-grained adaptive batching methodology SimiGrad that supports swift batch size adaption at the iteration level. The core component of SimiGrad is a novel lightweight gradient variance measurement that can capture the gradient variance change per iteration without expensive analysis. Our adaptive batching algorithm uses the gradient variance measurement to guide the batch size configuration so that the batch size is maximized while the gradient variance is maintained at a steady level. SimiGrad also supports automated learning rate scheduling and requires little manual effort. Finally, we integrate SimiGrad into mainstream machine learning frameworks with optimized real system performance and open-source it for easy adoption.

### 4.1  Measure Gradient Variance via Cosine Similarity of Batching

Existing works typically estimate gradient variance via collecting gradients with different batch sizes [16], which, however, is quite expensive due to the calculation of multiple batches as well as the process of collecting gradients from different batch sizes. In this paper, we propose a new way to measure the gradient variance by computing the cosine similarity of the gradient of a mini-batch.

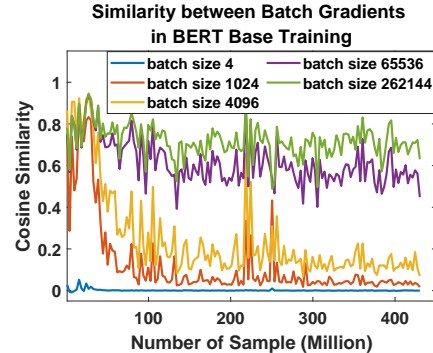

Figure 3: Cosine similarity in BERT pretrain.

Specifically, let $G = (g_1, g_2, ..., g_m)$ be the gradients of a batch of size $\frac{b}{2}$. The individual gradient $g_1, g_2, ..., g_m$ is assumed i.i.d. given a large enough training set [30]. Denote $G1 = (g1_1, g1_2, ..., g1_m)$ and $G2 = (g2_1, g2_2, ..., g2_m)$ as the aggregated gradients collected by SimiGrad, which are two independent observations from the distribution of $G$. The cosine similarity of aggregated gradients $G1, G2$:

$$cos(G1, G2) = \frac{G1 \cdot G2}{||G1|| * ||G2||}, \tag{1}$$

$$G1 \cdot G2 = \sum_{i=1}^{m} g1_i g2_i = \frac{1}{4} \sum_{i=1}^{m} (g1_i + g2_i)^2 - \frac{1}{4}(g1_i - g2_i)^2. \tag{2}$$

Taking the expectation on both sides, we have

$$\mathbb{E}(G1 \cdot G2) = \frac{1}{4}\sum_{i=1}^{m}\mathbb{E}(g1_i + g2_i)^2 - \frac{1}{4}\sum_{i=1}^{m}\mathbb{E}(g1_i - g2_i)^2. \tag{3}$$

For a random variable $X$, we have $Var(X) = \mathbb{E}(X^2) - [\mathbb{E}(X)]^2$. Using this formula, we rewrite the above equation as follows:

$$\mathbb{E}(G1 \cdot G2) = \frac{1}{4}[\sum_{i=1}^{m}Var(g1_i + g2_i) + \sum_{i=1}^{m}[\mathbb{E}(g1_i + g2_i)]^2 - \sum_{i=1}^{m}Var(g1_i - g2_i) - \sum_{i=1}^{m}[\mathbb{E}(g1_i - g2_i)]^2]. \tag{4}$$

As $g1, g2$ are assumed to follow the same distribution, we have $Var(g1_i + g2_i) = Var(g1_i - g2_i)$ and $\mathbb{E}(g1_i - g2_i) = 0$. Thus, we have

$$\mathbb{E}(G1 \cdot G2) = \frac{1}{4}\sum_{i=1}^{m}[\mathbb{E}(g1_i + g2_i)]^2 = \sum_{i=1}^{m}[\mathbb{E}(g_i)]^2, \tag{5}$$

$$\mathbb{E}(cos(G1 \cdot G2)) = \frac{\mathbb{E}(G1 \cdot G2)}{\mathbb{E}(||G1|| * ||G2||)} = \frac{\sum_{i=1}^{m}[\mathbb{E}(g_i)]^2}{\mathbb{E}(\sum_{i=1}^{m}(g_i)^2)}. \tag{6}$$

We can derive the relationship between the trace of covariance (total variance) $Tr(Cov(G))$ of gradients and the cosine similarity measurement $cos(G1, G2)$ proposed in SimiGrad as follows:

$$Tr(Cov(G)) = \sum_{i=1}^{m}Var(g_i) = \sum_{i=1}^{m}E(g_i^2) - \sum_{i=1}^{m}[E(g_i)]^2 = (1 - E(cos(G1, G2)))\sum_{i=1}^{m}E(g_i^2). \tag{7}$$

Note that the $Tr(Cov(G))$ are computed with half batch size $\frac{b}{2}$. To get a relationship between trace of covariance $Tr(Cov(G_{full}))$ on full batch size and the cosine similarity, we could use the conclusion from [30] to scale the variance as

$$Tr(Cov(G_{full})) = \frac{n-b}{2n-b}Tr(Cov(G)) = \frac{n-b}{2n-b}(1 - E(cos(G1, G2)))\sum_{i=1}^{m}E(g_i^2), \tag{8}$$

where $n$ is the total data samples and $b$ is the batch size.

Thus, the cosine similarity is closely related to the gradient covariance. As the cosine similarity does not require computing additional results using different batch sizes, our approach allows to evaluate the gradient variance at each iteration with little overhead.

At each iteration, the cosine similarity is computed using 2 aggregated gradients, where we utilize the replicas in distributed training systems. A replica is a local copy of model and optimizer on certain hardware (typically a GPU) [31, 32]. As shown in Fig.1, in each training step, all replicas $r_1, r_2, ..., r_n$ start with the same weights $\theta$ but with different mini-batches. After the feed forward and backward propagation of loss, each replica has its own gradient information $G_1, G_2, ..., G_n$. Then allreduce, a collective operation [33], is called to aggregate gradients from replicas, thanks to its excellent capability of utilizing network bandwidth to achieve fast communication.

Before the gradients are aggregated by allreduce, we split replicas equally into two groups, denoted as $\{r_1, r_3, ...\}$ and $\{r_2, r_4, ...\}$. We perform aggregation using allreduce in each group. This process is visualized in Fig. 2. The aggregated gradients in each group are denoted as $G1 = \frac{2}{n}\sum_{i=1}^{\frac{n}{2}}G_{2i-1}$ and $G2 = \frac{2}{n}\sum_{i=1}^{\frac{n}{2}}G_{2i}$, respectively. These are equivalent to the gradients of two mini-batches with half of the batch size. Then we compute the cosine similarity of the two group gradients as the measurement of gradient variance $\Phi = cos(G1, G2)$. Once the gradient variance is computed, we aggregate the two group gradients again using allreduce to compute the final aggregated gradients. At this point, we reach the step as the normal workflow.

When computing the cosine similarity, the training pipeline is not interrupted and the main overhead is an additional allreduce call and a cosine similarity computation. This is a significant cost reduction, compared with the gradient variance evaluation in previous works that need to collect multiple additional batches with different batch sizes or use heuristic algorithms. Taking BERT-Large pretrain on a 16 GPU cluster as an example, the overhead of an extra allreduce call and cosine similarity

computation is only around 10-15 ms, which is irrelevant to the batch sizes. In comparison, the gradient computation of a single 8k batch of BERT takes about 441ms and the time grows with the batch size and the need of computing multiple batch sizes [16].

To demonstrate the proposed cosine similarity is an effective measure for tracking the gradient variance, we plot the cosine similarity over the BERT pretraining for different batch sizes in Fig. 3. It is observed that the similarity of the gradients increases with the batch size. For the small batch size of 4, due to the high "noise" in the gradients, the cosine similarity is almost always 0. For the batch sizes of 1024 and 4096, the gradients are stable initially. However, the gradient quickly becomes unstable and thus the cosine similarity drops. For the large batch sizes of 65536 and 262144, the cosine similarity is relatively stable over the time, which corresponds to the stable gradients we get.

## 4.2 Automated and Fine-grained Adaptive Batching Algorithm

Using the cosine similarity, we propose an automated and fine-grained adaptive batching algorithm to derive optimized large batch sizes. The total batch size is denoted as $B = m * s * d$, where $m$ is the micro batch size on each replica, $s$ is the gradient accumulation steps, and $d$ is the number of replicas. In practice, the number of replicas $d$ depends on the number of available GPUs and the micro batch size $m$ is limited by GPU memory. Thus the factor $d$ is given and $m$ has an upper bound $m_{\max}$.

Algorithm 1 provides an overview of our adaptive batching algorithm. For every $p$ steps, we compare the measured gradient similarity $\Phi$ with the user defined similarity threshold $\gamma$ and update the target batch size $B$ accordingly using a predefined incremental step (e.g., 10%) to make the batch gradient similarity as close to the threshold as possible. Then, Algorithm 2 is invoked to update the micro batch size $m$ and the gradient accumulation steps $s$ to make the effective batch size as close to the target batch size as possible. Based on the updated batch size, we update the learning rate accordingly.

Note that in Algorithm 1, the only required parameter is the similarity threshold $\gamma$ and the other parameters are optional to improve the flexibility of the algorithm. Specifically, the adjust interval $p$ is used when training on small batch sizes or slow networks to reduce overhead. The batch size lower bound $B_{\min}$ is introduced for special scenarios where a well-tuned learning rate scheduling is provided. The batch size upper bound $B_{\max}$ is used when training with only limited epochs to ensure a reasonable number of updates to the model. In Sec. 5.3.2, we perform an ablation study to analyze the impact of the parameters in Algorithm 1. The results indicate that Algorithm 1 performs well with a broad range of $\gamma$ even without any optional parameters.

---

**Algorithm 1:** Adaptive Batching Algorithm Overview

---

**User specifications:** similarity threshold $\gamma$, adjust interval $p$ (optional), batch size lower bound $B_{\min}$ (optional), batch size upper bound $B_{\max}$ (optional), micro batch size upper bound $m_{\max}$ (optional).
**Initialization:** learning rate modifier $\alpha = 1$. If $m_{\max}$ not set, $m_{\max} = m$. If $p$ not set, $p = 1$.
**For every $p$ steps in training**
1) Retrieve similarity measurement $\Phi$.
2) If $\Phi < \gamma$, target batch size $B = 0.9B$, else, $B = 1.1B$.
3) Update the batch size to approximate $B$ based on Algorithm 2.
4) Update the learning rate modifier $\alpha$ according to $\frac{\alpha_{new}}{\alpha_{old}} = \sqrt{\frac{B_{new}}{B_{old}}}$.

---

---

**Algorithm 2:** Adaptive Batch Size

---

**Input:** Target batch size $B$, micro batch size upper bound $m_{\max}$, batch size lower bound $B_{\min}$ (optional), batch size upper bound $B_{\max}$ (optional). (Here, '\' means integer division)
1) If $B_{\max}$ set, bound the batch size $B = \min(B, B_{\max})$.
2) If $B_{\min}$ set, bound the batch size $B = \max(B, B_{\min})$ .
3) If $m_{\max}$ set, update micro batch size $m = \min(B \backslash d, m_{\max})$.
4) Update gradient accumulation steps $s = B/m \backslash d$.

---

## 4.3 Learning Rate Scheduling Support

Existing works [10, 11, 12, 26, 24, 13] suggest that the learning rate should be adjusted while adapting batch sizes in order to achieve good performance. We employ a learning rate scaling rule from [10, 24, 19] to adjust the learning rate automatically using a square root factor of the batch size

change (see Step 4 of Algorithm 1). It is worth noting that SimiGrad works with or without warmup [13]. If warmup is not used (i.e., a random initial learning rate is used), SimiGrad can still achieve good performance. It is interesting to observe that the learning rate automatically adjusted by the learning rate scaling rule resembles a similar pattern as the fine-tuned warmup scheduling (see Fig. 5). This indicates SimiGrad is robust and provides good support for the learning rate adaption.

## 4.4 Machine Learning System Integration

We can further optimize SimiGrad runtime performance by instrumenting the allreduce operation. An allreduce process can be decomposed into: a reduce-scatter process, an allreduce among groups, and an allgather process inside each group. After the reduce-scatter process, each replica holds one portion of the reduced gradients in each group. Thus we can directly compute the cosine similarity from the gradients. Then the allreduce is executed among groups but only to exchange the portion of the reduced gradients. Here the transferred data volume size is inversely proportional to the degree of parallelism and thus the overhead is small. Finally, the allgather is executed inside each group. In this way, we can avoid the overhead of an extra allreduce to further reduce the overhead of SimiGrad.

Modern machine learning frameworks are built with fixed micro batch sizes. Even if we can instrument the framework to make it support flexible batch sizes, it may disrupt the optimized existing training pipelines, such as the widely used NVIDIA DALI[34]. Therefore, we opt to design a batch dispatch queue that accumulates the data samples until the micro batch size is reached before aggregating them into a micro batch to be executed. This general design enables our approach to be easily integrated into almost any machine learning framework.

Furthermore, we do not directly change the learning rate as the client scripts may operate or depend on it (e.g., a custom learning rate scheduler). Instead, we use a learning rate modifier factor to indicate the learning rate scale with the adaptive batch size, which is applied during model weights update.

# 5 Evaluation

We evaluate SimiGrad via extensive experiments using popular benchmarks and demonstrate the superiority of SimiGrad over these benchmarks using the following criteria:

- *Performance*: we compare the performance of SimiGrad with state-of-the-art adaptive batching and large batch training techniques to demonstrate superior results on ResNet18, ResNet50, and BERT.
- *Generalization*: we demonstrate that SimiGrad establishes a better Pareto frontier of average batch size under different evaluation metrics, and sustains good generalization at very large batch sizes.
- *Robustness*: we show that SimiGrad does not need a manual learning rate warm-up schedule required by most large batch training techniques or careful tuning of hyperparameters, which makes SimiGrad robust against hyper-parameter variations.
- *Scalability*: we demonstrate that SimiGrad allows deep learning training to scale more efficiently on a wide range of GPU clusters with both slow and fast interconnect.

## 5.1 Evaluation Setup

Evaluations are performed on Ubuntu 18.04 on NVIDIA DGX-2 nodes (16 V100 GPUs). The applications are evaluated with DeepSpeed 0.2, Pytorch 1.6+, and CUDA 10.1 and 11.1. To ensure fair comparison, the hyperparameters, including learning rate and seeds, are kept the same between the baseline and our approach evaluations. Gradient accumulation is used when the batch size exceeds the graphic memory limit. For SimiGrad, the optional adjust interval $p$ is 1 by default for all evaluations and the gradient accumulation steps $s$ is adjusted during the training using Algorithm 2.

- *ResNet18 on CIFAR10*: The training lasts 200 epochs with SGD optimizer. The initial learning rate is 0.1, which decays at epoch 80 and 120 by factor 0.1, respectively. For SimiGrad, we set the similarity threshold 0.1 and the maximum batch size 2048.
- *ResNet50 on ImageNet*: The training lasts 90 epochs with SGD optimizer. The initial learning rate is 0.1, which decays at epoch 30, 60, and 80 by factor 0.1, respectively. For SimiGrad, we set the similarity threshold 0.1 and the maximum batch size 32k.
- *BERT Pretrain on WiKi and BookCorpus Datasets*: We pretrain BERT Large [2] from scratch for 170 epochs on WiKi and BookCorpus datasets with fixed sequence length 128 and 512 using LAMB optimizer. The training procedure follows the recommendation from the BERT paper [2], based on which we train epoch 1-150 using the sequence length 128 and epoch 151-170 using the sequence length 512. We evaluate it for two cases: (i) a well-tuned warmup learning rate scheduling is used and (ii) a simple learning rate decay is used. For the warmup case, the learning rate follows an exponential increase to 0.011 at epoch 9 and then decays with a factor of 0.9 for every 250 steps until epoch 150. For the simple case, a fixed learning rate 0.0006 with a linear decay is used until epoch 150. For both cases, the sequence length 512 training follows a warmup scheduling

| ResNet18 on CIFAR10 (SGD) | | | | |
| --- | --- | --- | --- | --- |
| | Number of Updates | Average Batch Size | Number of Samples | Test Accuracy (%) |
| Baseline | 78125 | 128 | 10M | 94.28 |
| AdaBatch | 15137 | 330 | 5M | 94.38 |
| AdaptDL | 7489 | 1335 | 10M | 93.07 |
| **SimiGrad** ($\gamma$=0.1, $b_{max}$=2048) | **5094** | **1963** | **10M** | **94.33** |
| ABSA | 4327 | 2311 | 10M | 85.85 |
| ResNet50 on ImageNet (SGD) | | | | |
| | Number of Updates | Average Batch Size | Number of Samples | Test Accuracy (%) |
| Baseline | 56250 | 2048 | 115M | 76.56 |
| Hand-tuned Baseline | 7038 | 16384 | 115M | 75.3 |
| **SimiGrad** ($\gamma$=0.1, $b_{min}$=2k, $b_{max}$=32k) | **23284** | **4952** | **115M** | **76.85** |
| **SimiGrad** ($\gamma$=0.4, $b_{min}$=2k, $b_{max}$=32k) | **7007** | **16456** | **115M** | **75.71** |
| BERT Large Pre-Training | | | | |
| | Number of Updates | Average Batch Size | Number of Samples | SQuAD F1 score |
| BERT Paper Baseline | 1000000 | 256 | 256M | 90.9 |
| LAMB Paper Baseline | 8599 | 59542 | 512M | 90.58 |
| NoWarmup Baseline | 106793 | 4096 | 437M | 86.9 |
| **SimiGrad with warmup** ($\gamma$=0.5, $b_{min}$=64k) | **5620** | **77831** | **437M** | **90.69** |
| **SimiGrad with warmup** ($\gamma$=0.8, $b_{min}$=64k) | **3425** | **127711** | **437M** | **89.82** |
| **SimiGrad without warmup** ($\gamma$=0.7, $b_{max}$=128k) | **6604** | **66236** | **437M** | **90.87** |
| **SimiGrad without warmup** ($\gamma$=0.8) | **4779** | **91530** | **437M** | **90.39** |
| **SimiGrad without warmup** ($\gamma$=0.6) | **7207** | **60694** | **437M** | **90.52** |

until learning rate 0.02 and then decay with a factor of 0.9 for every 150 steps. The batch size for sequence length 512 is fixed at 32k.

**State-of-the-Art Adaptive Batching Methods.** We cross-compare the performance of SimiGrad with other approaches that can adaptively adjust the batch size during training as well as the hand-tuned baselines. The evaluation metrics are the average batch size and the final generalization performance. The approaches we compared with are:

- *AdaBatch[28]*: a hard tuned strategy that gradually increases the batch size $b$ and the learning rate $\alpha$ along the training to keep $b/\alpha$ constant.
- *AdaptDL*: a library that adaptively sets the batch size based on the gradient noise scale [16] and sets the learning rate according to AdaScale [14].
- *ABSA[15]*: an adaptive batching scheme using second order methods and adversarial training.

### 5.2 Performance Analysis

For CIFAR-10, Table 1 indicates that SimiGrad achieves a higher test accuracy than the baseline, while keeping the largest average batch size among the other approaches that have similar test accuracies. Compared with AdaBatch that uses 15k model updates, SimiGrad uses only 5k model updates, which indicates almost 3x speedup in terms of the number of updates.

For ImageNet, Table 1 shows that SimiGrad achieves a better test accuracy with twice as large as the baseline batch size. Note that we use the vanilla SGD without tricks (such as label-smoothing and mixup data augmentation) and sophisticated learning rate scaling (such as LARS). Under this setup, the state-of-the-art approach [35] achieves 75.3% test accuracy with 16k batch size under well-tuned learning rate scaling, warmup, and decay. To show the performance of SimiGrad with larger batch

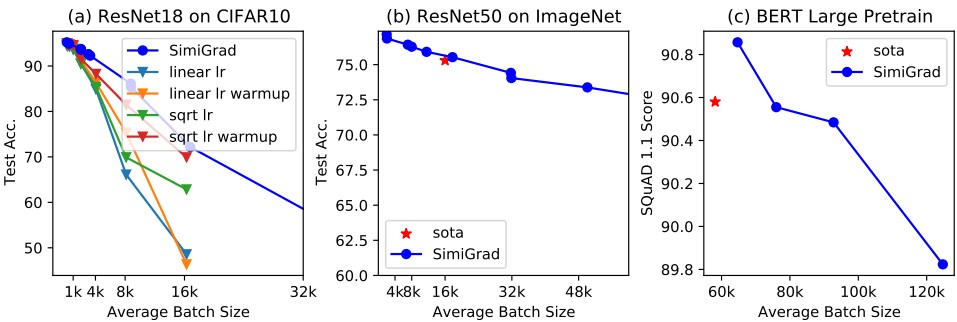

Figure 4: Pareto frontier of the average batch size under different performance metrics. For the same average batch size, SimiGrad always performs better than the other benchmarks. 'sota' (state-of-the-art) in (b) is from [35] and 'sota' in (c) is from [19].

sizes, we test more aggressive settings with similarity threshold 0.4, minimum batch size 2k, and maximum batch size 32k. Under this setting, the average batch size of SimiGrad reaches 16k, while keeping the test accuracy at 75.71% which is 0.4% higher than the hand-tuned results in [35].

For BERT Large pretraining, the model has 340 million parameters. The results in Table 1 shows the advantage of SimiGrad in large scale training over other approaches (the original BERT paper and the state-of-the-art LAMB). Specifically, even with an ultra large batch size 128k (using similarity threshold 0.8), SimiGrad can achieve comparable SQuAD scores. With less extreme batch size 66k (using similarity threshold 0.5), SimiGrad can achieve a better SQuAD score 90.85 than the state-of-the-art LAMB that uses an average batch size 59k and achieves SQuAD score 90.58.

## 5.3 Generalization and Robustness

To demonstrate the generalization and robustness of SimiGrad with large batch sizes, we generate the Pareto frontier of the average batch size under different performance metrics for different applications by grid search. As illustrated in Fig. 4, it can be observed that the model performance degrades as the batch size increases, which agrees with the observations in related work. It is worth noting that SimiGrad is able to keep the model performance reasonably good for a wide range of batch sizes and constantly outperforms the state-of-the-art approaches.

### 5.3.1 Robustness under Learning Rate Schedules

Learning rate warmup and learning rate scaling are two common techniques used to improve the model generalization for large batch training. Fig. 4(a) shows the impact of two common learning rate scaling rules (the linear scaling and squared root scaling) with large batch size; for both scenarios, we also show the impact of learning rate warm up. As illustrated in Fig. 4(a), the results with warmup have better test accuracies than the counterpart without warmup, and the squared root scaling is better than the linear scaling. However, despite the effectiveness of the learning rate warmup and the learning rate scaling, identifying an optimal schedule that works well for a certain batch size requires a non-trivial tuning effort.

Notably, SimiGrad can mitigate this effort and outperform even well tuned schedules with minimal tuning. To show the robustness of SimiGrad with untuned learning rate, we evaluate SimiGrad's performance for BERT pretrain without using any warmup but with a *untuned* learning rate with linear decay (see the details in Sec. 5.1). The results are presented in Table 1. Compared with the scenario of well-tuned learning rate, SimiGrad with untuned learning rate chooses more conservative batch sizes (still larger than state-of-the-art batch sizes) but has better SQuAD scores. These results also outperform the state-of-the-art results. Compared with SimiGrad, NoWarmup Baseline in Table 1 (the same untuned learning rate with LAMB without any warmup schedule using a fixed batch size 4k) shows a non-trivial performance degradation.

The reason why SimiGrad is robust against the learning rate warm-up is because it uses the gradient similarity metric to automatically adjust the effective learning rate at run time. From Fig. 5(a), it can be observed that even when no learning rate warmup schedule is provided, SimiGrad is able to come up with a warmup-like learning rate scaling based on the adaptive batch size, which not only saves considerable tuning efforts but also leads to a better model performance than the hand-tuned warmup schedules. When highly tuned warmup schedule is available for a large batch size, SimiGrad can leverage it to start the training from a pre-tuned large batch size, and increase it further over the training, which ultimately reaches a larger average batch size.

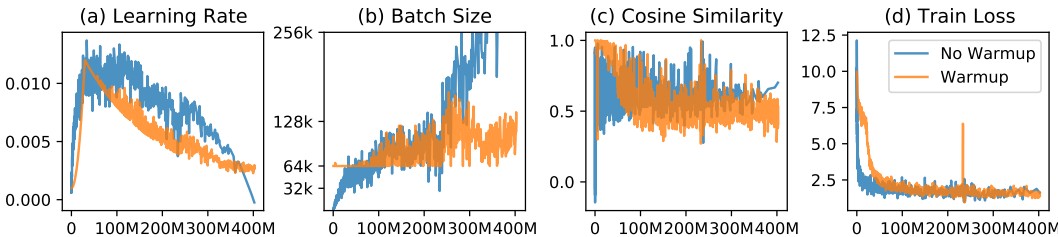

Figure 5: Details of BERT Large pretrain with adaptive batch size in sequence length 128.

### 5.3.2 Robustness Ablation Study

We perform the ablation study to evaluate SimiGrad's robustness against different parameters. First, we get rid of all optional parameters (such as batch size bounds) and extra learning rate scheduling (warmup), and run the BERT pretrain with the gradient similarity $\gamma = 0.6$. The results are shown in the bottom of Table 1, which shows comparable performance as the other experiments and outperforms the baselines. This indicates that SimiGrad can work well without any optional parameters nor extra learning rate scheduling. Then, in the same setting we evaluate the impact of gradient similarity values $\gamma = [0.6, 0.7, 0.8]$ on the performance of SimiGrad. The corresponding SQuAD scores are 90.52, 90.22, and 90.38 with the average batch size 60k, 75k, and 91k, respectively. This further indicates that SimiGrad can perform well under different gradient similarity thresholds.

We also evaluate how the adjustment interval impacts SimiGrad performance for ResNet18 on CIFAR-10. The adjustment intervals we tested are 5,10,100,200,400. The corresponding number of updates are 6053, 6456, 9333, 11299, 15732 and the test accuracy are 94.27, 94.23, 94.34, 94.39, 94.36. The ablation study results suggests that the scalability (batch size) gets worse as the adjustment interval grows while the accuracy remains the state-of-the-art.

### 5.4 Real-world Scalability and Speed Up Indication

Large batch size enables training the model on a large number of GPUs without getting bottle-necked by the all-reduce communication. If we assume that a GPU can compute $X$ samples per second, then a batch size of $B$ can be processed by $N$ gpus in

$$iteration\_time = \frac{B}{NX} + allreduce\_cost = \frac{B}{NX} + 2 \times \frac{N-1}{N} \times \frac{M}{bw} \qquad (9)$$

where, the absolute allreduce cost can be written as function of $N$, network bandwidth ($bw$) between GPUs and the model size ($M$). But relatively, more time is spent in allreduce as $N$ gets larger or $B$ gets smaller. Therefore, a large batch size is necessary to achieve good efficiency.

Using (9), we can evaluate the potential impact of batch sizes on end-to-end training time as a function of GPU cluster size and GPU-GPU bandwidth within the cluster. We measure $X$ using an NVIDIA V100 GPU, and simulate the end-to-end training time for BERT-Pretraining on a small, medium and large cluster with slow (1GB/s), moderate (5GB/s)

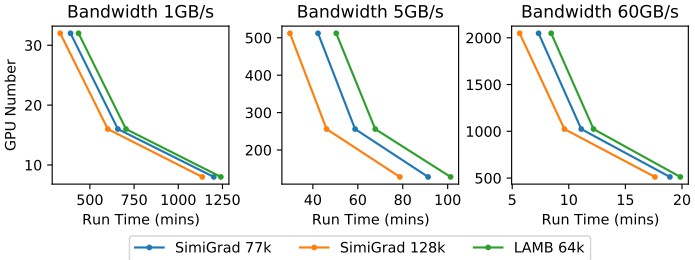

Figure 6: Cosine similarity of different batch sizes in BERT pretrain.

and fast interconnect (60 GB/s). Our simulation in Fig. 6 shows that SimiGrad achieves a 23.67%, 40.44%, and 32.97% reduction in training time compared to baseline on 32, 512 and 2048 GPUs on the small, medium and large clusters with slow, moderate and fast interconnect, respectively. This demonstrates the potential real world impact of our approach in improving training scalability across a wide range of actual GPU clusters with varying network bandwidth.

## 6 Conclusion

In this paper, we propose SimiGrad, a fine-grained adaptive batching methodology for enabling swift batch size adaption at the iteration level. The core component of SimiGrad is a lightweight gradient similarity measure based on cosine similarity and an enhanced synchronization approach. Our optimized machine learning system integration allows SimiGrad to be executed with negligible overhead in practice. SimiGrad achieves record breaking large batch size of 78k for BERT-Large pretraining with SQuAD score of 90.69.

## Acknowledgments and Disclosure of Funding

This work is supported in part by the National Science Foundation grants CAREER-2048044 and IIS-1838024. We thank the anonymous reviewers for their insightful comments.

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
