# OpenReview forum: "SimiGrad: Fine-Grained Adaptive Batching for Large Scale Training using Gradient Similarity Measurement"
_NeurIPS.cc/2021/Conference — NeurIPS 2021 Poster_

### Official Review · Reviewer_44BP · 2021-07-16

**Rating:** 6
**Confidence:** 3

**Summary:**

This paper study the scaling problem of large batch training.
This paper propose an adaptive method that can automatically adjust batch size to achieve good performance while scaling the training with large batch size.
Different from previous method that gather gradient variance information and adjust batch size at epoch level, the proposed method adjust the batch size at iteration level, which is more adaptive.

**Main Review:**

Strength:
- The idea is straight forward.
- The empirical performance is strong.

Weakness & Questions:
- I have a general question: It seems like the connection between bathsize and learning rate is strong [1]. Can the same intuition be applied to just adjusting learning rate? Such that, we can always fully utilize the hardware with large batch size.
- I don't see a clear connection between gradient variance and cosine similarity.
- Is SimiGrad sensitive to adjust interval $n$? The paper claim that iteration level batch-size adjustment is superior to epoch-level adjustment. An ablation study showing that a smaller $n$ leads to better performance is necessary.
- For CV tasks, the similarity threshold is 0.1, while it is 0.6 for BERT. I am curious what happens if  0.1 is used  for BERT.
- In line 312, does SimiGrad really achieve 3x speedup? The average batchsize of AdaBatch is merely 1/6 of SimiGrad. I expect the per-iteration speed of AdaBatch will be faster.

Minor:
- In line 142, where the data "441ms" comes from? which data/model?
- Figure 4. What is 'sota'

Reference:
[1] Don't Decay the Learning Rate, Increase the Batch Size

**Time Spent Reviewing:**

1

---

> ### Author Response · Authors · 2021-08-10
> **Response to Reviewer 44BP**
>
> Thanks for your encouraging and positive comments on our idea and results.
>
> Please find our response to all your concerns below:
>
> 1.	**Comment:** I have a general question: It seems like the connection between bathsize and learning rate is strong [1]. Can the same intuition be applied to just adjusting learning rate? Such that, we can always fully utilize the hardware with large batch size.
>
> - **Response:** [1] assumes a well-tuned learning rate scheduling is known so that it can be used to derive the corresponding batch size scheduling. However, in practice, a well-tuned learning rate scheduling is difficult and expensive to obtain, especially for large-scale training where the model training takes hundreds of GPU hours. Another issue is the scalability. Tuning batch size directly helps maximize the scalability. As shown in Figure 2 of [1], their method takes more than 25,000 updates to train a ResNet-style model on CIFAR10 whereas our approach only takes 5,094 updates. While a better hand-tuned learning rate scheduling may improve results to be comparable as SimiGrad, SimiGrad doesn’t require hand-tuned learning rate scheduling thus easier to use.
>
> 2.	**Comment:** don't see a clear connection between gradient variance and cosine similarity.
>
> - **Response:** We have attached the mathematical proof of the relationship between cosine similarity and gradient covariance below. We will add the proof in revision.
>
> Let $G=(g_1,g_2,...,g_m)$ be the gradients of a batch of size $\frac{b}{2}$. &nbsp; $g_1,g_2,...,g_m$ could be assumed as i.i.d as discussed in [2] given a large enough training set. Let $G1=(g1_1,g1_2,...,g1_m)$, $G2=(g2_1,g2_2,...,g2_m)$ be the aggregated gradients collected by SimiGrad, which are two independent observations from the distribution of $G$. The cosine similarity of aggregated gradients $G1, G2$ in SimiGrad is
> $$cos(G1,G2)=\frac{G1\cdot G2}{||G1||*||G2||}$$
> $$G1\cdot G2=\sum_{i=1}^{m} g1_ig2_i = \frac{1}{4} \sum_{i=1}^{m} (g1_i+g2_i)^2-\frac{1}{4} (g1_i-g2_i)^2$$
> Taking the expectation on both sides, we have
> $$\mathbb{E}(G1\cdot G2)=\frac{1}{4} \sum_{i=1}^{m}\mathbb{E}(g1_i+g2_i)^2-\frac{1}{4} \sum_{i=1}^{m}\mathbb{E}(g1_i-g2_i)^2$$
> For a random variable $X$, we have $Var(X)=\mathbb{E}(X^2 )-[\mathbb{E}(X)]^2$.
> Using this formula, we rewrite the above equation as follows:
> $$\mathbb{E}(G1\cdot G2)=\frac{1}{4}\[\sum_{i=1}^{m}Var(g1_i+g2_i)+\sum_{i=1}^{m}\[\mathbb{E}(g1_i+g2_i)\]^2-\sum_{i=1}^{m}Var(g1_i-g2_i)-\sum_{i=1}^{m}\[\mathbb{E}(g1_i-g2_i)\]^2\]$$
>
> As $g1,g2$ are assumed to follow the same distribution, we have $Var(g1_i+g2_i )=Var(g1_i-g2_i )$ and $\mathbb{E}(g1_i-g2_i )=0$.
> Thus, we have
>
> $$\mathbb{E}(G1\cdot G2)=\frac{1}{4}\sum_{i=1}^{m}[\mathbb{E}(g1_i+g2_i )]^2 =\sum_{i=1}^{m}[\mathbb{E}(g_i)]^2$$
> $$\mathbb{E}(cos(G1\cdot G2))=\frac{\mathbb{E}(G1\cdot G2)}{\mathbb{E}(||G1||*||G2||)}=\frac{\sum_{i=1}^{m}[\mathbb{E}(g_i)]^2}{\mathbb{E}(\sum_{i=1}^{m}(g_i)^2)}$$
>
> For the covariance, we can derive the relationship between the trace of covariance (total variance) $Tr(Cov(G))$ of gradients and the cosine similarity measurement $cos⁡(G1,G2)$ proposed in SimiGrad as follows:
> $$Tr(Cov(G))=\sum_{i=1}^{m}Var(g_i)=\sum_{i=1}^{m}E(g_i^2)-\sum_{i=1}^{m}\[E(g_i)\]^2=(1-E(cos(G1,G2)))\sum_{i=1}^{m}E(g_i^2)$$
>
> Note that the $Tr(Cov(G))$ are computed with half batch size $\frac{b}{2}$. To get a relationship between trace of covariance $Tr(Cov(G_{full}))$ on full batch size and the cosine similarity, we could use the conclusion from [1] to scale the variance as
> $$Tr(Cov(G_{full}))=\frac{n-b}{2n-b}Tr(Cov(G))=\frac{n-b}{2n-b}(1-E(cos(G1,G2)))\sum_{i=1}^{m}E(g_i^2)$$, where $n$ is the total data samples and $b$ is the batch size.
>
> We also empirically validate our conclusion. We found that by applying an exponential moving average to the gradient covariance as discussed in [2], the pattern of the cosine similarity and the trace of smoothed covariance are similar.
>
> 3.	**Comment:** Is SimiGrad sensitive to adjust interval ? The paper claim that iteration level batch-size adjustment is superior to epoch-level adjustment. An ablation study showing that a smaller n leads to better performance is necessary.
>
> - **Response:** We performed an ablation study over the adjustment interval for CIFAR10 under the same parameters {$\gamma$=0.1, ${b_max}$=2048}. Detailed training settings are described in Section 5.1. The adjustment intervals we tested are {5,10,100,200,400}. The corresponding number of updates are {6053, 6456, 9333, 11299, 15732} and the test accuracy are {94.27, 94.23, 94.34, 94.39, 94.36}. The ablation study results suggests that the scalability (batch size) gets worse as the adjustment interval grows while the accuracy remains the state-of-the-art. This verifies that the iteration-level adjustment is superior to the epoch-level adjustment.
>
> 4.	**Comment:** For CV tasks, the similarity threshold is 0.1, while it is 0.6 for BERT. I am curious what happens if 0.1 is used for BERT.
>
> - **Response:** We tried the similarity threshold 0.1 for BERT pretrain with 128 sequence length. The resulting average batch size is quite small, around 128, which suggests the intrinsic variance in BERT is smaller than that of CV tasks. We believe this is due to the fact that a single data sample in BERT training consists of many tokens (128 or 512 depends on the sequence length) which reduces the variance of gradients. Thus the sweet spot of similarity in BERT is higher than CV tasks.
>
> 5.	**Comment:** In line 312, does SimiGrad really achieve 3x speedup? The average batchsize of AdaBatch is merely 1/6 of SimiGrad. I expect the per-iteration speed of AdaBatch will be faster.
>
> - **Response:** When the training scale is large, e.g., with many GPUs, the per GPU batch size would be relatively small. For relatively small batch size, the per-iteration time is usually not linear to the batch size [3]. This is because the larger batch can take better advantage of the parallel threads of GPU due to the larger input dimension. In addition, the communication overhead can further dilute the difference in the computing time of per-iteration time between small and large batch size. For instance, in our profiling, a batch of size 1 takes 6.28ms per iteration whereas a batch of size 8 takes 6.79ms per iteration. Thus, despite of the average batch size of AdaBatch is merely 1/6 of SimiGrad, the per-iteration time could be only around 5~10% shorter. However, because SimiGrad only need to do about 1/3 of updates compared to AdaBatch, the overall speedup is around 3x in this case. We will clarify this in revision.
>
> 6.	**Comment:** In line 142, where the data "441ms" comes from? which data/model?
>
> - **Response:** It is from BERT pre-training on WikiandBookCorpus dataset. We will add more details in revision.
>
> 7.	**Comment:** Figure 4. What is 'sota'
>
> - **Response:** “SOTA” is an abbreviation of “state-of-the-art”. We will replace SOTA with state-of-the-art in revision.
>
> &nbsp;
>
> We will address all the above comments in revision. Thanks again for your valuable feedback.
>
> &nbsp;
>
> Related Work
>
>
> [1] (Ref. [22] in our submission) *Samuel L. Smith and Pieter-Jan Kindermans and Chris Ying and Quoc V. Le,* . "Don't Decay the Learning Rate, Increase the Batch Size." *6th International Conference on Learning Representations, ICLR 2018, Vancouver, BC, Canada, April 30 - May 3, 2018, Conference Track Proceedings. OpenReview.net,*
>
> [2] Wu, Jingfeng, et al. "On the noisy gradient descent that generalizes as sgd." International Conference on Machine Learning. PMLR, 2020. https://arxiv.org/pdf/1906.07405.pdf
>
> [3] You, Yang, Igor Gitman, and Boris Ginsburg. "Scaling sgd batch size to 32k for imagenet training." arXiv preprint arXiv:1708.03888 6 (2017): 12.

---

> > ### Comment · Reviewer_44BP · 2021-08-26
> > **Two more comments**
> >
> > The explanation of connecting gradient variance with cosine similarity looks good to me.
> > In Figure 4, I mean you should point out what is the SOTA method, i.e., adding a citation.

---

> > > ### Author Response · Authors · 2021-08-26
> > > **Response 2 to Reviewer 44BP**
> > >
> > > Thanks again for your valuable comments. The SOTA data point in Figure 4 is from [1]. We will be specific about the SOTA method by adding citations.
> > >
> > > [1] (Ref. [34] in our submission) Chris Ying and Sameer Kumar and Dehao Chen and Tao Wang and Youlong Cheng, . "Image Classification at Supercomputer Scale". CoRR abs/1811.06992. (2018).

---

### Official Review · Reviewer_mR3m · 2021-07-17

**Rating:** 7
**Confidence:** 3

**Summary:**

Large batch sizes are required to accelerate the large-scale training of deep learning (DL) models, which often comes at the cost of generalization performance. This paper introduces SimiGrad, an automated method that adapts the batch size at the mini-batch level during DL training. The proposed method achieves state-of-the-art performance on BERT-Large when employing very large batch sizes.

The first contribution of the paper is the use of the cosine similarity metric to measure the gradient variance of the gradient of a mini-batch. This replaces the typical expensive step that needs to calculate multiple batches and collect gradients from different batch sizes.

SimiGrad also automatically optimizes the learning rate of the training procedure. The automated adaptation of the learning rate used by SimiGrad is shown to out-perform other learning rate scaling rules when using large batch sizes.

SimiGrad is integrated into existing ML systems and the experimental results show that state-of-the-art results can be obtained when SimiGrad is applied to BERT-Large training using a record-breaking batch size of 77K.

**Limitations And Societal Impact:**

The authors do not discuss the potential negative societal impact of their work, but that looks reasonable to me. I do not see any negative impact of their proposed work.

In terms of technical limitations of the work, the authors analyze the robustness of SimiGrad to the mandatory gradient similarity hyper-parameter. The authors could further discuss the impact of the optional hyper-parameters on the performance of their scheme.


**Main Review:**

In terms of significance, the paper addresses an important and very-well studied problem, namely the optimization of the batch size and learning rate in DL training. It is shown that the proposed method can achieve state-of-the-art results when employing very large batch size. More precisely, the test accuracy of the BERT-Large model is improved over the state-of-the-art accuracy by using 77K batch size.

In terms of originality, the paper uses a simple but effective and novel idea to measure the gradient variance using cosine similarity. Moreover, it introduces an algorithm that automatically adapts the batch size and the learning rate during training. In contrast to many approaches from the state-of-the-art, the learning rate is not adapted using a linear or a squared scaling rules. The learning rate is computed based on the ratio between the new and the old batch size.

The paper is fairly easy to read and includes a comprehensive related work section and a clear explanation of the motivation and challenges of the addressed problem. The clarity and quality could be improved as suggested in the following paragraphs.

- Figure 1 should show that the similarity is used to update the weights of the model.
- In section 4.2, what are the `n` steps in line 217? Is `n` related to `s`? The similarity threshold is a hyper-parameter of the proposed methodology. How is it tuned and how it relates to `n`?
- For consistency with the description of Algorithm 1, the authors should use `B` instead of `b` in Algorithm 2.
- When mentioning modern ML frameworks, please exemplify and provide references. otherwise it is too generic.
- How many nodes have been used for running the experiments, how many GPUs per node?
- How were the similarity threshold and the maximum batch size chosen for the experiments.
- It would be good if the authors would include information about the training times of the different methods. This would allow a good accuracy-time trade-off analysis. Is the time model proposed in (1) validated by the experiments?
- In Table 1, why only 5M samples were used for training AdaBatch? Moreover, why not writing SimiGrad instead of `Ours` in the BERT sub-table.
- What are the integration challenges when implementing the proposed solution into existing ML frameworks?
- How can the model in (1) be extended to support hierarchical scenarios? For examples, two nodes each with X GPUs, the local GPUs are interconnected via a higher bandwidth than the inter-node connection bandwidth.
- Figure 5 shows that with no warmup the batch size explodes to sizes larger than 256K, why this difference between warmup and no warmup?

**Time Spent Reviewing:**

6

---

> ### Author Response · Authors · 2021-08-10
> **Response to Reviewer mR3m**
>
> Thanks for your encouraging and positive comments on the problem importance, solution innovation, study thoroughness, and state-of-the-art results of our work.
>
> Please find our response to all your concerns below:
>
> 1.	**Comment:** Figure 1 should show that the similarity is used to update the weights of the model.
>
> - **Response:** We will update Fig. 1 as suggested.
>
> 2.	**Comment:** In section 4.2, what are the n steps in line 217? Is n related to s? The similarity threshold is a hyper-parameter of the proposed methodology. How is it tuned and how it relates to n?
>
> - **Response:** There is a typo in our submission, where we used the same symbol $n$ for both the number of replicas and the optional adjust interval. We will fix it. The optional adjust interval is irrelevant to gradient accumulation steps and we observe no correlation between it and similarity threshold. The adjust interval is designed to add flexibility for ultra-scale training like GPT-3 where one may want to control the frequency of adjustment.
>
> - The similarity threshold is treated as a hyper-parameter and is currently hand-tuned by employing a bisection approach. As similarity threshold is a bounded value between 0 and 1, the tuning overhead is relatively low. We plan to explore more efficient tuning approach for similarity threshold in our future work.
>
> 3.	**Comment:** For consistency with the description of Algorithm 1, the authors should use B instead of b in Algorithm 2.
>
> - **Response:** We will update Algorithm 2 as suggested.
>
> 4.	**Comment:** When mentioning modern ML frameworks, please exemplify and provide references. otherwise it is too generic.
>
> - **Response:** We implement and evaluate SimiGrad on DeepSpeed, which is a Pytorch-based open-source framework. Our implementation and all evaluation scripts could be found via the anonymous link in footnote 1 on page 2 of our submission.
>
> 5.	**Comment:** How many nodes have been used for running the experiments, how many GPUs per node?
>
> - **Response:** We use 1 or 2 NVIDIA DGX-2 machines for most of the evaluation. Each node has 16 V100 GPUs each with 32GB VRAM.
>
> 6.	**Comment:** How were the similarity threshold and the maximum batch size chosen for the experiments.
>
> - **Response:** The similarity threshold is treated as a hyper-parameter and is currently hand-tuned by employing a bisection approach. As similarity threshold is a bounded value between 0 and 1, the tuning overhead is relatively low. We plan to explore more efficient tuning approach for similarity threshold in our future work.
> - The maximum batch size is set to ensure scaling rules hold, i.e., batch size needs to be much smaller than the total number of samples [1].
>
> 7.	**Comment:** It would be good if the authors would include information about the training times of the different methods. This would allow a good accuracy-time trade-off analysis. Is the time model proposed in (1) validated by the experiments?
>
> - **Response:** The GPU cluster we used for our experiments is not a completely isolated environment thus the wall-clock training time is difficult to be measured accurately. In addition, the training time is GPU cluster dependent. Considering the above, we follow literate [2][3][4] to use the “number of updates” and “average batch size” as the main measurements since they are GPU cluster independent and can be accurately measured and reproduced. To validate the real-world scalability and speed up, we performed simulation of the batch size impact on training time and reported the results in Figure 6 of Section 5.4. Model (1) is validated by experiments.
>
> 8.	**Comment:** In Table 1, why only 5M samples were used for training AdaBatch? Moreover, why not writing SimiGrad instead of Ours in the BERT sub-table.
>
> - **Response:** AdaBatch is a hand-tuned batch size scheduling that is designed to work only on 5M data samples. We tried to extend it to 10M but observed no improvement in the test accuracy. Thus we choose to use AdaBatch’s original setting of 5M. The Table 1 will be updated as suggested.
>
> 9.	**Comment:** What are the integration challenges when implementing the proposed solution into existing ML frameworks?
>
> - **Response:** One key engineering challenge is to instrument the all-reduce operation to limit it within two groups of replicas. We addressed this challenge by using the approach discussed in Section 4.1. Another challenge is to reduce the memory and runtime overhead and we discussed the solution in Section 4.4.
>
> 10.	**Comment:** How can the model in (1) be extended to support hierarchical scenarios? For examples, two nodes each with X GPUs, the local GPUs are interconnected via a higher bandwidth than the inter-node connection bandwidth.
>
> - **Response:** The general term $bw$ in our model (1) is the measured bandwidth from tools like nccl-test. Therefore, the effects of different communication architecture and optimization have been reflected in this measurement already.
>
> 11.	**Comment:** Figure 5 shows that with no warmup the batch size explodes to sizes larger than 256K, why this difference between warmup and no warmup?
>
> - **Response:** Figure 5 demonstrates the training details of some entries in Table 1. The warmup case is corresponding to the row 19 of Table 1 where {$\gamma$=0.5, $b_{min}$=64k} is used. The no warmup case is corresponding to row 23 in Table 1 where {$\gamma$=0.6} is used. We can see that the similarity threshold obtained by bisection search is different from the no warmup case (0.6) to the warmup up case (0.5). Given such different similarity target, the adjusted batch size of the no warmup case is much larger compared to the warmup. We plan to further investigate the reasons behind the adjustment behaviors in our future work.
>
> &nbsp;
>
> We will address all the above comments in revision. Thanks again for your valuable feedback.
>
> &nbsp;
>
> Related Work:
>
> [1] (Ref. [22] in our submission)
> *Samuel L. Smith and Pieter-Jan Kindermans and Chris Ying and Quoc V. Le,* . "Don't Decay the Learning Rate, Increase the Batch Size." *6th International Conference on Learning Representations, ICLR 2018, Vancouver, BC, Canada, April 30 - May 3, 2018, Conference Track Proceedings. OpenReview.net,*
>
> [2] (Ref. [10] in our submission)
> *Elad Hoffer and Itay Hubara and Daniel Soudry,*. "Train longer, generalize better: closing the generalization gap in large batch training of neural networks." *Advances in Neural Information Processing Systems 30: Annual Conference on Neural Information Processing Systems 2017, December 4-9, 2017, Long Beach, CA, USA.*
>
> [3] (Ref. [11] in our submission)
> *Christopher J. Shallue and Jaehoon Lee and Joseph M. Antognini and Jascha Sohl-Dickstein and Roy Frostig and George E. Dahl, .* "Measuring the Effects of Data Parallelism on Neural Network Training". *J. Mach. Learn. Res. 20. (2019): 112:1–112:49.*
>
> [4] (Ref. [12] in our submission) *Masters, Dominic, and Carlo Luschi.* "Revisiting small batch training for deep neural networks." *arXiv preprint arXiv:1804.07612 (2018).*

---

### Official Review · Reviewer_xSCv · 2021-07-18

**Rating:** 7
**Confidence:** 4

**Summary:**


This paper addresses the issue of parallel, large mini-batch SGD, and how to adapt learning rate  and batch size. The aim is to find a sweet spot with batches large enough reduce gradient estimator variance while not too large, with incremental gains in gradient variance reduction, and potentially harming generalization.  Specifically, they consider a fast way to compute the gradient variance.  The issue is that computing the covariance for the gradient estimator is very expensive, especially if done so with different batch sizes (as is done, eg in [16]).  While prior methods for adapting LR or batch size have computed the gradient variance relatively infrequently, e.g., every epoch, this paper argues that gradient variance can vary significantly on finer time scales, and hence they aim to estimate gradient variance and adapt batch at every mini-batch.  The experiments in the paper are extensive, and they show promising results with extremely large batch sizes.



**Limitations And Societal Impact:**

Given the cost of training large models, work like this addressing faster convergence through batch size adaptation is important.


**Main Review:**

I enjoyed reading this paper.  The topic will be of interest to a large fraction of the NeurIPS community, and the innovations and experiments in the paper appear promising.

I do have several concerns and suggestions that the authors may wish to address.

1) Section 4.1 begins 'Existing works typically estimate gradient variance via collecting gradients with different batch sizes'.  I think it might be worth explaining that for IID data, the estimator variance for the mean should decreases like 1/N multiplied by the variance of the RV. So from a single batch one can in estimate the gradient covariance. A noisy estimate of the mean can, however be problematic, in which case it is sometimes effective to use two samples.  This is explained in Appendix A.1 in [16] for example (ie https://arxiv.org/pdf/1812.06162.pdf).

Indeed, there are strong similarities between the approach proposed in the current paper and in reference [16].  In particular, Appendix A.1 in [16] appears to describe a method similar to SimiGrad that they say is "free in a data-parallel environment".  It outlines a similar idea to SimiGrad that uses averages before and after allreduce.  This I think a more detailed discussion of similarities and differences with [16] may be useful.

2) The relation between normalized gradients, cosine similarity, and the gradient covariance is not well explained in the paper, but should be.  This is a major omission, but it can be corrected reasonably straightforwardly.  In particular, one can show that the cosine similarity in this case is affinely related to the trace of the covariance matrix for the normalized gradient vector.  In this sense, cosine similarity is closely related to the total variance of the normalized gradients in the batch.  Explaining this relation between cosine similarity of the aggregated gradients, and the trace of the covariance, and the assumptions under which it holds, will take a few steps to explain clearly, but I believe it will greatly help to clarify and motivate SimiGrad.

That said, I note that a proper estimate of the total variance (or trace of the covariance) is not strictly necessary as SimiGrad requires the user to specify a target value of the cosine similarity (gamma in the paper).  One only requires a quantity that is related to some measure of the variance, up to a constant for example, since one only compares this quantity to a threshold whose units are not critical.

3) It would be useful to clarify the hardware used for the experiments in the paper.  In particular, what size of GPUs do the workers use, and how much VRAM is available.  More details about the parameters used in the experiments would also be appreciated.  Section 5.1 states that the maximum batch size is 2K for CIFAR10, 32K for ImageNet, and 32K for text data with sequence length 512 when training BERT.  The maximum batch size for data with a sequence length of 128 is not specified.  The issue here is that the average batch size in Figure 4 goes far beyond the maximum batch sizes listed.  Perhaps it would be helpful to explicitly specify experiment parameters values in terms of the variables used in the algorithms (ie b_max, m_max, n, and s, etc).

I also have a few minor points:
1) In several places the paper uses the term variance where the correct quantity of interest is the covariance.

2) pg 9, line 391: Given that the experiment discussed just before the conclusion section is in simulation, I don't think the paper should claim a demonstration of real-world impact, but rather potential real-world impact.

3) pg 4, line 190: should be g_i in the summations instead of r_i.



**Time Spent Reviewing:**

4 hours

---

> ### Author Response · Authors · 2021-08-10
> **Response to Reviewer xSCv**
>
> Thanks for your encouraging and positive comments on the topic, innovation, and experiments of our work.
>
> Please find our response to all your concerns below:
>
> 1. **Comment:** Section 4.1 begins 'Existing works typically estimate gradient variance via collecting gradients with different batch sizes'. I think it might be worth explaining that for IID data, the estimator variance for the mean should decreases like 1/N multiplied by the variance of the RV. So from a single batch one can in estimate the gradient covariance. A noisy estimate of the mean can, however be problematic, in which case it is sometimes effective to use two samples. This is explained in Appendix A.1 in [1] for example (ie https://arxiv.org/pdf/1812.06162.pdf).
> Indeed, there are strong similarities between the approach proposed in the current paper and in reference [1]. In particular, Appendix A.1 in [1] appears to describe a method similar to SimiGrad that they say is "free in a data-parallel environment". It outlines a similar idea to SimiGrad that uses averages before and after allreduce. This I think a more detailed discussion of similarities and differences with [1] may be useful.
>
> - **Response:** We will update the description in Section 4.1 to add the discussions of estimator variance.
>
> - Yes, both the approach in [1] and SimiGrad aim to optimize the batch size by tracking the gradient noise, but the tracking methods are quite different.
> (1) the implementation described in A.1 of [1] highly depends on the parallelism degree of the system whereas SimiGrad supports flexible parallelism degree. (2) The measurement proposed in A.1 of [1] is noisy and needs to use a hand-tuned exponential decay parameter to stabilize it.  (3) [1] assumes the learning rate is well-tuned and close enough to optimal whereas SimiGrad has no assumptions on learning rate. In fact, we have implemented the approach in [1] and we found that their noise level is quite high when using in a common cluster of 16 GPUs. We also found that their noise level decreases as the number of GPUs increases, which indicate their design highly depends on the parallelism degree. In comparison, SimiGrad can derive reasonably stable measurements in a parameter free and system independent manner as shown in Fig. 3 of our submission. In summary, SimiGrad is a more general and easier to use approach, compare with [1].
>
>
> 2. **Comment:** The relation between normalized gradients, cosine similarity, and the gradient covariance is not well explained in the paper, but should be. This is a major omission, but it can be corrected reasonably straightforwardly. In particular, one can show that the cosine similarity in this case is affinely related to the trace of the covariance matrix for the normalized gradient vector. In this sense, cosine similarity is closely related to the total variance of the normalized gradients in the batch. Explaining this relation between cosine similarity of the aggregated gradients, and the trace of the covariance, and the assumptions under which it holds, will take a few steps to explain clearly, but I believe it will greatly help to clarify and motivate SimiGrad.
> That said, I note that a proper estimate of the total variance (or trace of the covariance) is not strictly necessary as SimiGrad requires the user to specify a target value of the cosine similarity (gamma in the paper). One only requires a quantity that is related to some measure of the variance, up to a constant for example, since one only compares this quantity to a threshold whose units are not critical.
>
> - **Response:** We have attached the mathematical proof of the relationship between cosine similarity and gradient covariance below. We will add the proof in revision.
>
> Let $G=(g_1,g_2,...,g_m)$ be the gradients of a batch of size $\frac{b}{2}$. &nbsp; $g_1,g_2,...,g_m$ could be assumed as i.i.d as discussed in [2] given a large enough training set. Let $G1=(g1_1,g1_2,...,g1_m)$, $G2=(g2_1,g2_2,...,g2_m)$ be the aggregated gradients collected by SimiGrad, which are two independent observations from the distribution of $G$. The cosine similarity of aggregated gradients $G1, G2$ in SimiGrad is
> $$cos(G1,G2)=\frac{G1\cdot G2}{||G1||*||G2||}$$
> $$G1\cdot G2=\sum_{i=1}^{m} g1_ig2_i = \frac{1}{4} \sum_{i=1}^{m} (g1_i+g2_i)^2-\frac{1}{4} (g1_i-g2_i)^2$$
> Taking the expectation on both sides, we have
> $$\mathbb{E}(G1\cdot G2)=\frac{1}{4} \sum_{i=1}^{m}\mathbb{E}(g1_i+g2_i)^2-\frac{1}{4} \sum_{i=1}^{m}\mathbb{E}(g1_i-g2_i)^2$$
> For a random variable $X$, we have $Var(X)=\mathbb{E}(X^2 )-[\mathbb{E}(X)]^2$.
> Using this formula, we rewrite the above equation as follows:
> $$\mathbb{E}(G1\cdot G2)=\frac{1}{4}\[\sum_{i=1}^{m}Var(g1_i+g2_i)+\sum_{i=1}^{m}\[\mathbb{E}(g1_i+g2_i)\]^2-\sum_{i=1}^{m}Var(g1_i-g2_i)-\sum_{i=1}^{m}\[\mathbb{E}(g1_i-g2_i)\]^2\]$$
>
> As $g1,g2$ are assumed to follow the same distribution, we have $Var(g1_i+g2_i )=Var(g1_i-g2_i )$ and $\mathbb{E}(g1_i-g2_i )=0$.
> Thus, we have
>
> $$\mathbb{E}(G1\cdot G2)=\frac{1}{4}\sum_{i=1}^{m}[\mathbb{E}(g1_i+g2_i )]^2 =\sum_{i=1}^{m}[\mathbb{E}(g_i)]^2$$
> $$\mathbb{E}(cos(G1\cdot G2))=\frac{\mathbb{E}(G1\cdot G2)}{\mathbb{E}(||G1||*||G2||)}=\frac{\sum_{i=1}^{m}[\mathbb{E}(g_i)]^2}{\mathbb{E}(\sum_{i=1}^{m}(g_i)^2)}$$
>
> For the covariance, we can derive the relationship between the trace of covariance (total variance) $Tr(Cov(G))$ of gradients and the cosine similarity measurement $cos⁡(G1,G2)$ proposed in SimiGrad as follows:
> $$Tr(Cov(G))=\sum_{i=1}^{m}Var(g_i)=\sum_{i=1}^{m}E(g_i^2)-\sum_{i=1}^{m}\[E(g_i)\]^2=(1-E(cos(G1,G2)))\sum_{i=1}^{m}E(g_i^2)$$
>
> Note that the $Tr(Cov(G))$ are computed with half batch size $\frac{b}{2}$. To get a relationship between trace of covariance $Tr(Cov(G_{full}))$ on full batch size and the cosine similarity, we could use the conclusion from [1] to scale the variance as
> $$Tr(Cov(G_{full}))=\frac{n-b}{2n-b}Tr(Cov(G))=\frac{n-b}{2n-b}(1-E(cos(G1,G2)))\sum_{i=1}^{m}E(g_i^2)$$, where $n$ is the total data samples and $b$ is the batch size.
>
> We also empirically validate our conclusion. We found that by applying an exponential moving average to the gradient covariance as discussed in [2], the pattern of the cosine similarity and the trace of smoothed covariance are similar.
>
> 3.	**Comment:** It would be useful to clarify the hardware used for the experiments in the paper. In particular, what size of GPUs do the workers use, and how much VRAM is available. More details about the parameters used in the experiments would also be appreciated. Section 5.1 states that the maximum batch size is 2K for CIFAR10, 32K for ImageNet, and 32K for text data with sequence length 512 when training BERT. The maximum batch size for data with a sequence length of 128 is not specified. The issue here is that the average batch size in Figure 4 goes far beyond the maximum batch sizes listed. Perhaps it would be helpful to explicitly specify experiment parameters values in terms of the variables used in the algorithms (ie b_max, m_max, n, and s, etc).
>
> - **Response:** We use one or two NVIDIA DGX-2 nodes for evaluation. Each node has 16 V100 GPUs each with 32GB VRAM and 2 Intel(R) Xeon(R) Platinum 8168 CPU. The system memory for each node is 1.5TB. The GPUs are equipped with NVLink and nodes are interconnected by InfiniBand.
>
> - We use a broad range of parameters to evaluate SimiGrad. The optional adjust interval n is 1 by default for all evaluations and the gradient accumulation steps s is adjusted during the training using Algorithm 2. For CIFAR10, our parameters are {$\gamma$=0.1, $b_{max}$=2048} (row 6 in Table 1). For ImageNet, our parameters are {$\gamma$=0.1, $b_{min}$=2k, $b_{max}$=32k} (row 12 in Table 1) and {$\gamma$=0.4, $b_{min}$=2k, $b_{max}$=32k} (row 13 in table 1). For BERT, we use {$\gamma$=0.5, $b_{min}$=64k} (row 19 in Table 1), {$\gamma$=0.8, $b_{min}$=64k} (row 20 in Table 1), {$\gamma$=0.7, $b_{max}$=128k} (row 21 in Table 1), and {$\gamma$=0.8} (row 22 in Table 1), {$\gamma$=0.6} (row 23 in Table 1). Note that $b_{min}$ and $b_{max}$ are optional parameters.
> For Figure 4, we run a coarse-grained grid search over $\gamma$ with different $b_{max}$ (or no $b_{max}$).
>
> 4.	**Comment:** (1) In several places the paper uses the term variance where the correct quantity of interest is the covariance. (2) pg 9, line 391: Given that the experiment discussed just before the conclusion section is in simulation, I don't think the paper should claim a demonstration of real-world impact, but rather potential real-world impact. (3) pg 4, line 190: should be g_i in the summations instead of r_i.
> - **Response:** We will address them in the paper.
>
> &nbsp;
>
> We will add the above discussion, proof, and clarification in revision. Thanks again for your valuable feedback.
>
> &nbsp;
>
>
> Related work:
>
> [1]  (Ref. [16] in our submission) *McCandlish, Sam, et al.* "An empirical model of large-batch training." *arXiv preprint arXiv:1812.06162 (2018).* https://arxiv.org/pdf/1812.06162.pdf
>
> [2] *Wu, Jingfeng, et al.* "On the noisy gradient descent that generalizes as sgd." *International Conference on Machine Learning. PMLR, 2020.*
> https://arxiv.org/pdf/1906.07405.pdf

---

### Decision · Program_Chairs · 2021-09-27

**Decision:**

Accept (Poster)

**Comment:**

All reviewers like the simplicity of the proposed approach and think this paper makes a good contribution towards large batch training of DNNs. Reviewers also raised a number of concerns about methodology and comparisons. Authors have clarified some of them in the response. Overall I think the paper makes some interesting contributions and suggest acceptance. I encourage authors to update the final version addressing all the comments form the reviewers.